# Analysis of the Volatile Organic Compound Fingerprint of Greek Grape Marc Spirits of Various Origins and Traditional Production Styles

Maria Marinaki [1,2,3], Ioannis Sampsonidis [3,4], Alexandros Nakas [5,6], Panagiotis Arapitsas [7,8,*], Andreana N. Assimopoulou [5,6] and Georgios Theodoridis [1,2,3,*]

1. Department of Chemistry, Aristotle University of Thessaloniki, 54124 Thessaloniki, Greece; mmarinaki@chem.auth.gr
2. Biomic_Auth, Bioanalysis and Omics Laboratory, CIRI-AUTH, 57001 Thessaloniki, Greece
3. FoodOmicsGR_Research Infrastructure, Auth Node, CIRI-AUTH, 57001 Thessaloniki, Greece; isampson@nutr.teithe.gr
4. Department of Nutritional Sciences & Dietetics, International Hellenic University, 57001 Thessaloniki, Greece
5. Laboratory of Organic Chemistry, School of Chemical Engineering, Aristotle University of Thessaloniki, 54124 Thessaloniki, Greece; anakas@cheng.auth.gr (A.N.); adreana@cheng.auth.gr (A.N.A.)
6. Natural Products Research Centre of Excellence (NatPro-AUTh), CIRI-AUTh, 57001 Thessaloniki, Greece
7. Department of Wine, Vine and Beverage Sciences, School of Food Science, University of West Attica, 12243 Athens, Greece
8. Research and Innovation Centre, Fondazione Edmund Mach, 38010 Trento, Italy
* Correspondence: panagiotis.arapitsas@fmach.it (P.A.); gtheodor@chem.auth.gr (G.T.)

**Abstract:** The most well-known traditional Greek grape marc distillate made from winemaking pomace is called "Tsipouro". Its production involves various grape pomace cultivars, preparation protocols, and anise-flavoring or not, and it should be a colorless liquid with intense organoleptic properties due to the raw materials used in its production and have a minimum alcoholic strength of 37.5% by volume. This study aimed to characterize the volatilome of tsipouro products by covering as many geographical areas and production styles as possible, as there is a lack of characterization of the aromatic composition of this Greek traditional alcoholic beverage. A Headspace Solid Phase Microextraction Gas Chromatography–Mass Spectrometry (HS-SPME-GC-MS) method was applied in 60 samples, resulting in the identification and semi-quantification of over 90 volatile compounds. The statistical analysis pointed out the metabolites that characterized each traditional product group and underlined the influence of the geographical origin and the production protocol. Aniseed spirits from Northern Greece, Macedonia, Limnos Island, and Thessaly, produced from Muscat pomaces, were found to be richer in terpenes, terpenoids, and flavored compounds, attributing to product aroma and quality; different terpenoids were found to be dominant in Muscat distillates from different regions, showing the importance of geographical origin and production process. In conclusion, the results demonstrated the high aroma variability of the Greek Tsipouro, explained that this diversity is caused mainly by the raw material, and could be helpful in the better protection of the origin of this traditional product and the improvement of its quality.

**Keywords:** grape marc distillate; tsipouro; metabolomics; volatilome; HS-SPME-GC-MS; geographical region

## 1. Introduction

Greece has a long history of vineyard exploitation and is one of the world's top 20 wine-producing countries. Aside from winemaking, the production of Greek grape marc spirits is an important chapter in the Hellenic vine cultivation history [1,2]. "Tsipouro" is the most well-known traditional alcoholic grape marc distillate produced in Greece from winemaking residue (it is known as "tsikoudia" or "raki" on the island of Crete). According

to Issiodos, Hippocrates, and Aristotle, distillation in Greece has a long history and dates back to antiquity. Tsipouro and tsikoudia distillates are the outcomes of this tradition [1]. "Tsipouro" belongs to the "grape marc spirit" category and meets the relevant requirements set out in point 6 of Annex II to Regulation (EC) No 110/2008 [3] and the general requirements for "Tsipouro"/"Tsikoudia" as displayed in the technical documentation issued by the Directorate of Alcohol & Foodstuffs of the General Chemical State Laboratory of Greece [4]. "Tsipouro" is a transparent and colorless liquid with intensive organoleptic characteristics derived from the raw materials used (grape marc and perhaps sediment) and the specific methods of production (distillation means and processes). Its content in volatiles is equal to, or exceeding, 140 g per hectoliter of pure alcohol, while methanol does not exceed 1000 g per hectoliter of pure alcohol. The minimum alcohol by volume is 37.5% vol., and the addition of alcohol is not permitted.

The production process of these grape marc distillates includes the selection of the appropriate grape variety, thorough fermentation, and finally the slow and gradual distillation of the fermented grape marc [5,6]. Anise, fennel, saffron, and walnut tree leaves are a few aromatic seeds or plants that some viticulturists prefer to add. These spirits have recently been recognized as products of protected geographical designations when produced by professional distillers in accordance with European and national law in specific viticultural regions (Annex III, Regulation [EC] No. 110/2008) [3,7–9]. Similar distillates with equivalent appellations are produced in numerous countries, particularly around the Mediterranean Sea, including "zivania" in Cyprus, "grappa" in Italy, "aguardente" or "orujos" in Spain, and "bagaceira" in Portugal. They are also produced in central Europe, specifically "eau-de-vie de marc" in France, as well as in some other nations, such as "rakija" in Slavic nations and "tshiatshia" in Georgia [8,10].

Tsipouro is a well-known distillate in Greece, and includes different Protected Geographical Indications, such as "Tsipouro of Macedonia", "Tsipouro of Thessaly", and "Tsikoudia of Crete". Traditionally, in Northern Greece (Macedonia, Thessaly, Tyrnavos, North Aegean Islands) Tsipouro is often anise-flavored, while this practice is far less common in the South of the country. In Crete, grape marc spirits are called "Tsikoudia" and can also derived from raisin (Soultanina) [3,11]. The most common grape cultivars employed in the production of Tsipouro in Greece belong to the aromatic family of Muscat (e.g., Muscat in Macedonia and Thessaly, Muscat of Alexandria in Limnos), although other autochthonous or international cultivars are also used. Each region produces a grape marc spirit according to the tradition, the culinary heritage, the cultivated grapes, and the herds grown in the region. According to the General Chemical State Laboratory of Greece, in 2021, tsipouro exports reached an equivalent of 110 thousand liters of pure alcohol (lpa), showing a 37.5% increase in comparison to the previous year, and nearly a four-fold increase compared to 2012 abroad shipments [12]. Therefore, it is important to study the products of each region with a holistic approach in order to characterize the volatilome of tsipouro of different regions in Greece.

Several factors influence the volatile composition of grape marc spirits, including grape varietal origin, pomace storage method, fermentation duration, distillation technology, and the quality and length of the wooden barrels used in the aging process [13,14]. According to Regulation (EC) 787/2019, a grape marc spirit is required to have an ethanol content of at least 15% (*v/v*) [15,16]. Tsipouro, except for ethanol and water, is composed of hundreds of volatile and semi-volatile compounds derived from grape marc seeds and peels, or created during pomace fermentation [17]. These metabolites are responsible for organoleptic characteristics which determine the quality of the spirits. Furthermore, they are utilized as markers of grape variety and maturation state at harvest, geographical origin, grape pomace storage conditions, and/or distillation technique applied [18,19]. Aroma is a key factor in determining the organoleptic quality and style of a given distillate. Higher alcohols, acids, ethyl and acetate esters, aldehydes, and terpenes are the main group of compounds detected in alcoholic beverages. 2-Methylpropan-1-ol, amyl alcohols, 2-phenylethanol, acetic acid, acetaldehyde, ethyl acetate, ethyl lactate, 1-hexanol, linalool,

and *alpha*-terpineol are the major compounds responsible for the aroma of spirits such as tsipouro [1,7,9,15,20–22].

Several extraction techniques, analytical methods, and detectors have been applied by researchers for the study of the volatile composition of various grape marc spirits. Gas chromatography coupled with a flame ionization detector (GC-FID) [23–28] and gas chromatography coupled with mass spectrometry (GC-MS) with direct injection of the sample [29], commonly following an extraction procedure [10,18,30–32], have been previously used for the determination of volatile profile of European distillates. Only a few of these methods have been applied to study the volatilome of Greek tsipouro and tsikoudia; nevertheless, Nuclear Magnetic Resonance (NMR) spectroscopy approaches have been used to monitor the metabolic profile of Greek distillates [6,9,33].

Headspace Solid Phase Microextraction (HS-SPME) is a cutting-edge approach for analyzing aroma and volatile compounds because it is a simple, effective, and less expensive extraction process that requires limited solvent manipulation, and it is easily automated if a specific SPME autosampler is provided [7,34]. SPME is the most commonly used sample preparation procedure for the detection of volatile and semi-volatile metabolites in distillate samples. The main HS-SPME parameters, which affect the detection of several volatile compounds in distillates, are the type of SPME fiber, the extraction time and temperature, the desorption time and temperature, the sample volume, and the concentration of salt. SPME settings commonly entail the use of mixed-sorbent fibers and HS extraction at temperatures ranging from 25 to 55 °C for 15–60 min [13,33,35–37]. To the best of the authors' knowledge, whilst aroma is a quality attribute for alcoholic beverages, only one study previously applied an HS-SPME-GC-MS method to analyze Greek tsipouro [21].

This study aimed to determine the volatile metabolic fingerprint of Greek distillates tsipouro and tsikoudia in order to complete the lack of literature about the analysis of their volatilome using modern techniques. The aroma profile of tsipouro of different regions in Greece, produced by several native grape cultivars, was herein outlined for the first time.

## 2. Materials and Methods

### 2.1. Chemicals

4-Methylpentan-2-ol was used as an internal standard and was purchased from Merck (Darmstadt, Germany). Sodium chloride (NaCl) which was used in extraction for the salting-out effect, as well as methanol (MeOH), which was used for the dilution of internal standard, were purchased from Chem-Lab (Zedelgem, Belgium).

### 2.2. Samples

Sixty grape marc spirit samples produced in Greece and Cyprus in 2016 and 2018 were analyzed. Of these, 27 originated from Crete, 21 of which were mono- or multi-varietal spirits containing Soultanina, and 6 of them were produced from other native cultivars (Liatiko, Kotsifali, Mandilari); 20 were from Macedonia, 8 of them were produced in Thessaloniki, 4 in Pella, and the rest from various regions in Macedonia (Serres, Kilkis, Kavala, Litohoro, Nea Mesimvria); 9 were from Limnos Island, North Aegean; 3 were from Thessaly and 1 was from Cyprus. They were produced from several grape varieties with the addition of anise or not. More details of the samples are supplied in Table S1. The samples were kept in glass vials, at room temperature, in a shaded and dry place until they were analyzed.

### 2.3. HS-SPME Conditions and Sample Preparation

The desorption conditions and the concentrations of salt and fiber were selected according to the literature [13,18,31] as follows: the addition of 25% *w/v* sodium chloride (NaCl) was performed in the samples to control the ionic strength, 50/30 μm DVB/CAR/PDMS (divinylbenzene/carboxen/polydimethylsiloxane) SPME fiber was used, which is suitable for the detection of several volatiles from different compound groups, and a 10 min desorption at 230 °C was performed. A Box–Behnken Response Surface Methodology (RSM)

design was applied for the optimization of the extraction time and temperature, as well as sample volume, which were considered to be the most important factors [38]. Briefly, three different values for each factor were tested on pooled grape marc spirit samples that were representative of all samples. The optimal conditions resulting from the DoE were 45 °C extraction temperature, 35 min extraction time, and 4 mL sample volume.

Finally, 4 mL of sample, 1 g of NaCl for the salting-out effect, and 12 μL of a 1000 μg mL$^{-1}$ 4-methylpentan-2-ol internal standard solution in methanol, resulting in a final concentration of 3 μg mL$^{-1}$, were added in a 10 mL glass vial. The vials were sealed with a PTFE/silicone septum and were equilibrated at 45 °C for 10 min with agitation. The (DVB/CAR/PDMS) fiber (Sigma-Aldrich (Darmstadt, Germany), 2 cm length, 50/30 thickness) was then introduced to the headspace for 35 min at 45 °C. According to the manufacturer's recommendations, the fiber was conditioned at 270 °C.

A QC sample was prepared as a representative sample by mixing equal volumes of each wine sample and was injected during the study to evaluate the stability of the analytical system. Blank runs were also performed to reveal possible carryover. The samples were analyzed in randomized order. All grape marc spirits were analyzed with the aforementioned method to fingerprint the volatilome of each spirit and determine the differences between them according to their geographical origin. QC samples were analyzed at the beginning, after every 7 samples during sample analysis, and at the end of the batch of analyzed spirits to study the stability and reproducibility of the method. The identified compounds presented a relative standard deviation below thirty percent (RSD < 30%) and multivariate statistical models were applied to investigate the discrimination and correlation among samples.

### 2.4. Gas Chromatography–Mass Spectrometry

An EVOQ GC-TQ Bruker triple quadrupole (Bruker Daltonics, Bremen, Germany) system with a CTC-PAL autosampler (CTC Analytics AG, Zwingen, Switzerland) was used for the GC analysis. The chromatographic separation was carried out on an HP-INNOWAX (30 m × 0.25 mm × 0.25 μm) column (Agilent Technologies, Santa Clara, CA, USA). The oven temperature started at 50 °C, where it remained for 1 min, and was then increased at a 3 °C min$^{-1}$ rate to 70 °C, held there for 1 min, and then increased at a 3 °C min$^{-1}$ rate to 150 °C, and finally increased at a 5 °C min$^{-1}$ rate to 230 °C, where it remained for 4 min, resulting in a 55 min total runtime method. Helium (99.999%) as a carrier gas was set at a constant flow rate of 1 mL min$^{-1}$. Inlet temperature was set at 230 °C and the splitless injection mode was applied. Electron Ionization was applied at 70 eV and full scan spectra were acquired from 25 to 500 amu with a 250 ms scan time.

### 2.5. Data Processing and Chemometrics

Chromatographic data were treated by MSWS data processing software (Bruker Daltonics, Bremen, Germany) and NIST14 Mass Spectral Library. Mass Spectral Deconvolution and Identification System (AMDIS) along with the Assignment Validator and Integrator (GAVIN) script for MATLAB were used for chromatogram deconvolution and peak integration and identification, with a minimum match factor of 70%. The chromatographic peak areas of Extracted Ion Chromatograms (EICs) were used for the relative quantification by dividing the peak areas of the compounds of interest by the peak area of the internal standard (4-methylpentan-2-ol) and multiplying this ratio by the initial concentration of the internal standard (expressed as mg L$^{-1}$).

Multivariate statistical analysis was performed by using principal component analysis (PCA) and orthogonal projection to latent structures discriminant analysis (OPLS-DA) and validated by permutation plots and CV-ANOVA values. Statistical significance in the differences of the metabolites was evaluated by using Student's *t*-test and biomarker assessment via Variable Importance for the Projection (VIP) plots, S-plots, and p(corr) using the Soft independent modelling by class analogy (SIMCA) package (version 14.1;

Umetrics, Sweden) and the online platform Metaboanalyst (https://www.metaboanalyst. ca/, accessed on 26 May 2023).

## 3. Results and Discussion

### 3.1. Identification of Volatile Metabolites in Analyzed Grape Marc Spirits

Higher alcohols, acids, ethyl and acetate esters, carbonyl compounds, and terpenes were the major compound groups identified in analyzed distillates. A chromatogram of a QC grape marc spirit sample is illustrated in Figure 1. More than 90 metabolites were detected, identified, and semi-quantified with the proposed method, and their names, chemical formulae, molecular weights, and relative concentrations are presented in Table S2.

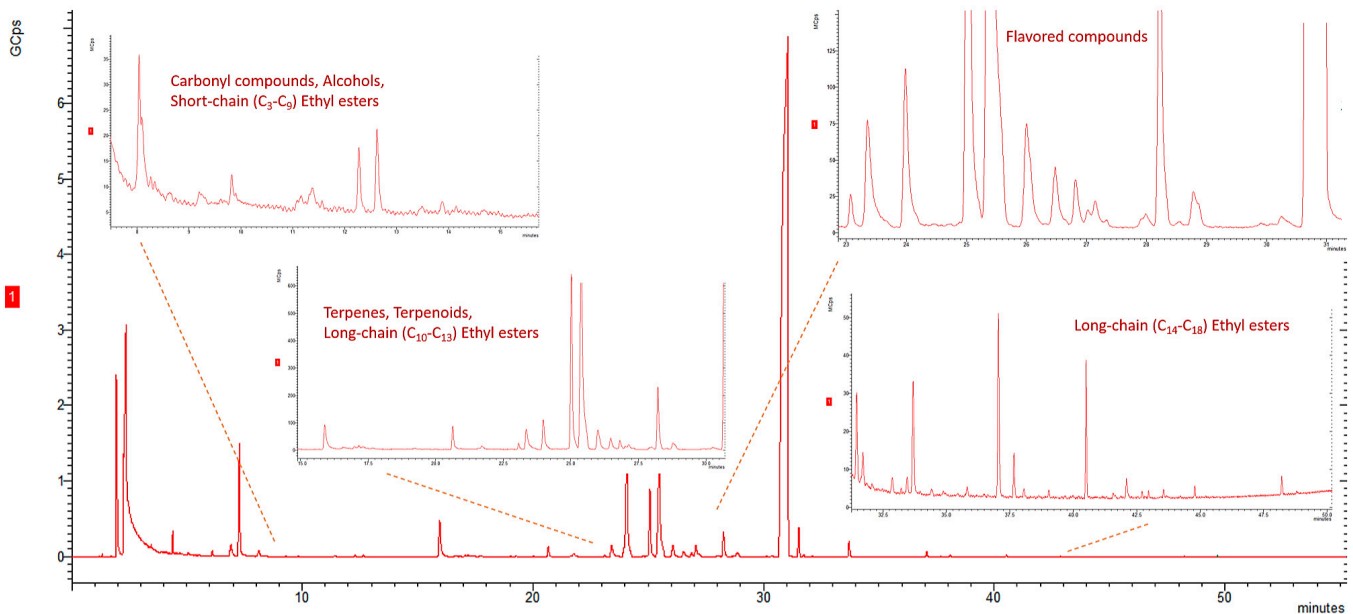

**Figure 1.** Representative HS-SPME-GC-MS chromatogram of volatile compounds identified in a QC grape marc spirit sample.

The volatile compound groups that are very important for the aromatic profile of Greek grape marc spirits are terpenes and terpenoids. Although higher concentrations of terpenes are expected in the initial fractions of distillation due to their good solubility in ethanol and poor solubility in water, their behavior during distillation is characterized by an increase in their concentration in the final fractions at higher temperatures because their relatively high boiling points are crucial in determining the distillation behavior [20]. The presence of monoterpenes in the aromatic profile of distillates is reflected by the grape variety from which the beverage is produced. Distillates produced by Muscat and Muscat of Alexandria grapes, such as Macedonia and Limnos spirits, have been found as richer in terpenes than spirits produced in other regions in Greece using different grape cultivars. Linalool, *alpha*-terpineol, nerol oxide, and geraniol were the major terpenoids identified in the analyzed samples. The concentration of the first two was 5-to-10-fold higher than the other terpenoids. These differences in concentrations have also been observed by several researchers. Lukic et al. [27] noticed that these terpenoids were also dominant in Muscat Ottonel distillates. Zocca et al. [39] detected them in Moscato and Prosecco grappas, while Stoica et al. [20] found that, in some cases under acidic conditions, nerol and geraniol were converted to linalool and *alpha*-terpineol. Linalool and nerol oxide were the major forms of terpenoids in Limnos and Thessaloniki samples, while the more stable forms of terpenoids, *alpha*-terpineol, and geraniol were found in higher levels in Thessaly and Thessaloniki samples, while terpinen-4-ol was the most abundant terpenoid in Thessaly and Macedonia samples, as illustrated in Figure 2. The concentration of hotrienol was found to be 10-fold

higher in Muscat of Alexandria grape marc samples from Limnos than any other analyzed distillate.

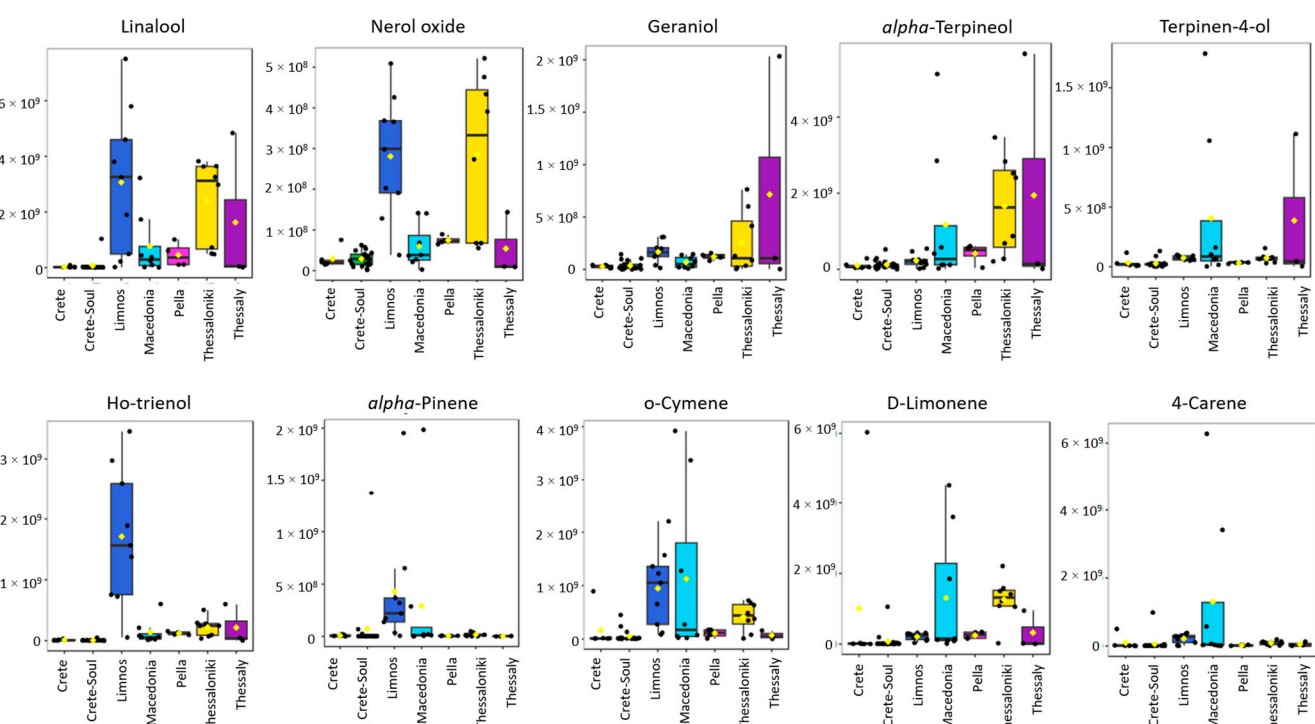

**Figure 2.** Boxplots of representative terpenoids among Greek grape marc spirit samples. Black dots represent the samples' values and yellow dots are the mean values.

*o*-Cymene [34,40,41], *alpha*-pinene [42–44], *beta*-myrcene [21], D-limonene [21,45], ylangene [18,29], *alpha*-cedrene [46] and *alpha*-terpinene [14,29] were detected mostly in Muscat, Muscat of Alexandria and Merlot wines, grape musts, and spirits, as they are characteristic metabolites of these cultivar grapes; however, in this study, their concentration was found to be over 50-fold higher in some Muscat grape marc spirit samples originated from Macedonia. Caryophyllene has been previously found in Cabernet Sauvignon berries [47], while Garcia-Martin et al. [35] found terpenols in orujo samples and Giannetti et al. [14] and Lukic et al. [19] detected terpenes, such as *alpha*-terpinene, *p*-cymene, *alpha*-cubebene, and *alpha*-calacorene in several European spirit samples.

The distillation of pressed fermented saccharated raw materials flavored with star aniseed (*Illicium verum*), fennel (*Foeniculum vulgare*), green aniseed (*Pimpinella anisum*), or other plants yields aniseed spirits. Anise (*Pimpinella anisum L.;* Umbelliferae family) has been cultivated throughout Europe and has been used as a popular aromatic herb and spice since antiquity. Anise is a plant native to the Middle East that has been used since ancient Egypt, and its major volatile compound is anethol, which is responsible for aniseed's fragrant characteristics [29,48,49]. Anethol was found in very high concentration in Limnos samples, followed by aniseed Muscat samples from Macedonia, while estragole, menthol, 4-anisaldehyde, *p*-anisaldehyde diethyl acetal, *alpha*-himachalene, *gamma*-himalachene, and aromadendrene, which were the main flavored compounds detected in aromatized Greek tsipouro, were found mostly in Limnos samples (Figure 3). Most of them have also been identified in Turkish raki, Greek and Cypriot ouzo, and other European distillates [29,50].

Aromatic terpenes, such as *alpha*-himachalene, *gamma*-himalachene, *alpha*-cadinene, and *alpha*-curcumene, have been previously identified in essential oils and aromatized distillates from Turkey, and their presence in Greek tsipouro arises from the addition of aromatic material during distillation [29,48,51]. 3-Acetylphenol, fenchone, carvone [52], and estragole were found in higher concentrations in Thessaly and Macedonia samples,

while *gamma*-isogeraniol, *trans-p*-mentha-1(7),8-dien-2-ol and thymyl ethyl ether were more abundant in Thessaly samples, as illustrated in Figure 4.

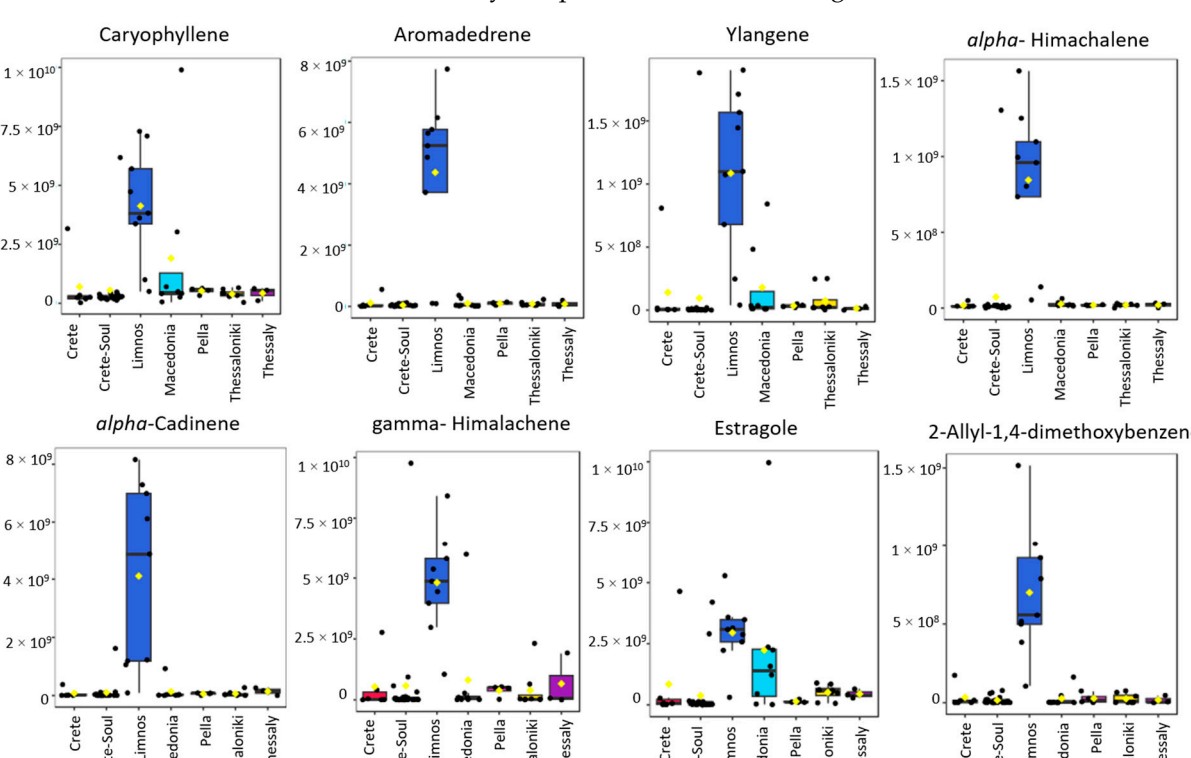

**Figure 3.** Boxplots of representative flavored compounds resulting from the addition of anise in Limnos tsipouro samples. Black dots represent the samples' values and yellow dots are the mean values.

According to Silva et al. [53], a group of volatile compounds that has a strong contribution to fruity and flowery aromas of distillates is ethyl esters. Their concentration is influenced by grape variety, but they are formed during the alcoholic fermentation of grape marc [25]. During the aging of distillates, the concentration of hexanoate, octanoate, and decanoate ethyl esters increases, reaching higher levels than that of their corresponding fatty acids. These ethyl esters were found to be more abundant in Macedonia, Thessaloniki, and Pella samples, while ethyl palmitate was higher in Limnos and Thessaloniki samples, as is shown in Figure S1, which are produced mainly from Muscat grapes, with the addition of anise, and red multi-variety grapes. Ethyl lactate is another important ester in grape pomace spirits, since it is considered to stabilize the spirit flavor and soften the harsh flavor characteristics [10]. Its presence in spirits in high concentrations, is linked to the pomace's malolactic fermentation by lactic bacteria, and the production of lactic acid by malic acid. Hence, the low concentration of this compound in the majority of the analyzed samples is indicative of the quality of the raw material and therefore the resulting distillate. These concentration levels are much lower than those of grappas and orujos found by Cortes et al. [25,30].

Ethyl acetate is often the most abundant ester of spirits, with an important contribution to the organoleptic characteristics of distilled alcoholic beverages, while concentration levels of 150–200 mg $L^{-1}$ can add spoilage notes to the wine, having "fingernail polish remover" properties [9,26,54]. The concentration of ethyl acetate in this study was found to be in low levels in most of the samples, which may contribute to the floral and fruity properties of the grape marc spirit. However, its concentration exceeded the aforementioned level in the distillates of Macedonia and Thessaly. Similar levels of ethyl acetate were detected by Apostolopoulou et al. [10] in Greek bottled and homemade tsipouro of Epirus. Methyl

acetate has also been detected in spirits in low concentrations. High concentration levels of these acetate esters are important indicators of aerobic fermentation and storage conditions, or an incorrect separation distillation fraction [9,30]. Isoamyl and phenethyl acetate were detected in all analyzed samples, contributing flowery and fruity notes to the beverages, but in lower concentrations than in traditional and industrial orujos samples [25].

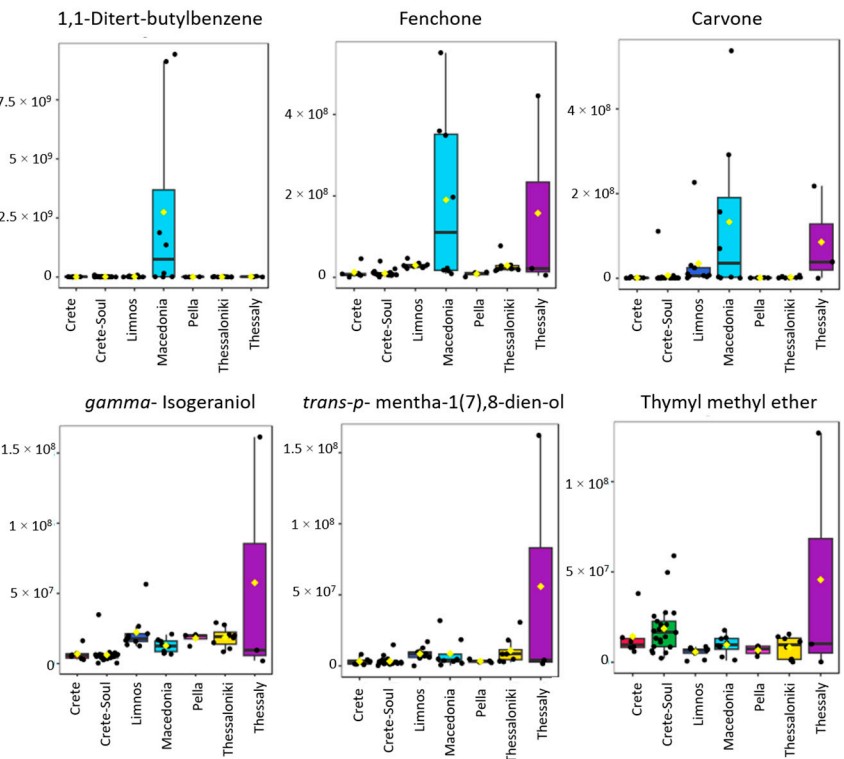

**Figure 4.** Boxplots of representative flavored compounds resulting from the addition of anise in Macedonia and Thessaly tsipouro samples. Black dots represent the samples' values and yellow dots are the mean values.

Acetic acid (acetic and vinegar-like odor) was the most abundant volatile acid in the analyzed distillates, with varying concentrations among samples, but under its perception threshold. Other studies have presented higher concentrations of acetic acid in European distillates, but in this work, samples were not fresh distillates, and, as is already known, the concentration of acids decreases after storage. Fatty acids have also been detected in grape marc spirits [55]. Octanoic and decanoic acids were the most important components of this compound group, with lower concentrations than acetic acid, being below 1 mg $L^{-1}$ in most cases. A high concentration of octanoic acid was detected in Thessaly samples, while *n*-decanoic acid was more abundant in Muscat and Muscat of Alexandria samples from Thessaly, Thessaloniki, and Limnos distillates at similar levels to Portuguese bagaceiras spirits [54] and Croatian Muscat Blanc and Muškat Ruža Porečki spirits [27].

A group of volatile compounds that is positively involved in the sensory quality of distillates is the group of higher alcohols. 2-Methylpropan-1-ol, amyl alcohols, and 2-phenylethanol were found to be the dominant alcohols in Greek distillates, as well as in other European beverages, such as grappas or orujos. The presence of amyl alcohols reinforces the structure of the distillates in the mouth, while 2-phenylethanol contributes to rosy, sweet, and perfume-like notes of the distillate [9,25,54]. 2-Methylpropan-1-ol and amyl alcohol levels increase during aging, and their concentrations were found as similar to other studies for Greek distillates [10,21]. 2-Phenylethanol tends to be distilled when the alcohol content is low, typically characterized as a tail product, so it should be present in marc distillates at low levels. However, this depends on both grape variety and distillation techniques employed [25,31]. Its concentration was higher in Thessaloniki and

some Macedonian samples produced from Muscat grapes; however, it did not present great differentiation among analyzed samples, as illustrated in Figure S2. It increases during the aging of wines and brandies, probably as a result of the transesterification reaction of phenethyl acetate [26]. 1-Hexanol is another alcohol contributing to the aroma of spirits, with positive herbaceous notes in low concentrations, or with coconut-like, harsh, and pungent aromas when the concentration exceeds 20 mg $L^{-1}$ [10,25,26,54]. An interesting observation is that the sample of Cyprus was very rich in higher alcohols, especially in isoamyl alcohol, which increased during aging.

Aldehydes, which are found in alcoholic beverages as a result of spontaneous or microbially mediated oxidation [26], were detected in tsipouro samples in low concentrations. Acetaldehyde, which is usually the major carbonyl compound in grape pomace distillates [25,56], was detected in much lower levels than its perception threshold (25 mg $L^{-1}$) and even lower than the concentration in the European distillates analyzed by Silva et al. [54] and Cortes et al. [25]. Whilst the concentration of acetaldehyde increases during aging, the levels of this metabolite in the analyzed samples suggest the high-quality of the Greek distillates. Moreover, this is in agreement with the results of Flouros et al. [26], who noticed that there was no statistically significant change in the concentration of acetaldehyde in Greek tsipouro after 12 months of storage. Acetaldehyde ethyl methyl acetal, acetaldehyde ethyl amyl acetal, and para-anisaldehyde diethyl acetal were observed to be more abundant in Macedonia and Thessaly tsipouro samples. On the other hand, the level of benzaldehyde in some samples shows that storage affected the formation of this metabolite, as it is associated with microbiological development during the storage process or the ensilage of grape pomace [30,35].

The formation of furfural in distillates is caused due to dehydration of fermentable sugars by heating in acidic conditions and/or Maillard reaction [9]. Its content is affected by the cultivar and excessive pressing of the corresponding marc [25]. Its odor is reminiscent of bitter almond and cinnamon and has a toxic character, but its perception threshold is 150 mg $L^{-1}$, 100-fold higher than the concentration in the analyzed samples. This concentration is closer to the concentration of furfural in grappa, which is produced using distillation equipment that does not imply direct heating on the mass of grape marc [25] compared with orujos or homemade tsipouro samples analyzed before [10]. An exception was the analyzed sample of Cyprus, which contained higher levels of benzaldehyde and furfural. This might be due to either aging or the traditional process of distillation. 2-Allyl-1,4-dimethoxybenzene was detected in 10-fold higher concentrations in Muscat of Alexandria of Limnos distillates, while the rest of the samples contained lower concentrations, as also reported by Dieguez et al. [44] in orujo spirits produced from several varieties.

### 3.2. Classification of Greek Distillates According to Geographical Region

Tsipouro samples contain hundreds of volatile and semi-volatile metabolites, which are indicators of geographic origin, and distillation techniques [18]. For example, Cortés et al. [25] found a lower content of volatile fermentation compounds (both qualitatively and quantitatively) in Italian spirits (Grappa) compared to Spanish Orujos, with 2-phenyl ethanol, hexenols, aldehydes, and most ethyl esters and acetates being responsible for the specific characteristics of these distillates. On the other hand, Kokoti et al. [21], who studied the effect of the distillation technique on the aromatic profile of tsipouro, found that the fractional column distillation gave the richest volatile profile, followed by home distillation and copper alembic distillation, mainly due to the high values of ethyl octanoate, ethyl decanoate, and DL-limonene, which were much higher when fractional column distillation was applied.

A PCA scores plot, presenting the differentiation of all analyzed samples in this study according to their geographical origin, is illustrated in Figure 5, and it is shown that distillates of Limnos were clearly differentiated from other samples, while samples from Crete produced from native grape cultivars were not discriminated from those produced from Soultanina cultivar. The three groups of Macedonian samples, Thessaloniki, Pella,

and the rest of Macedonia, presented a small separation, as they were all produced from Muscat pomaces and with the addition of anise, so OPLS-DA models were used for further investigation of the observed differences between production areas.

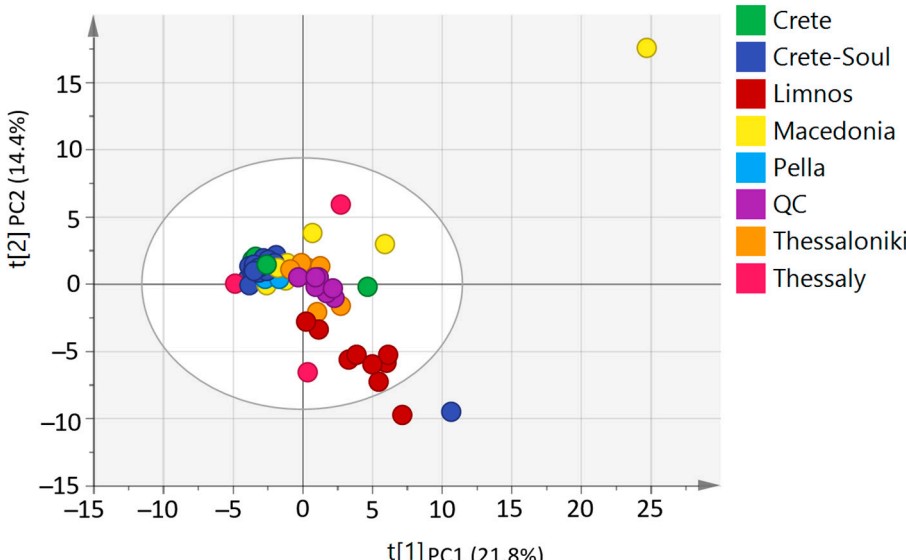

**Figure 5.** PCA scores plot presenting the discrimination of analyzed samples according to their geographical origin.

The OPLS-DA models were used to further study the differences among samples, and the statistical significance of each model was confirmed by permutations, Hotelling lines, and CV-ANOVA analysis ($p < 0.05$). Six statistically significant OPLS-DA models between the production areas of the analyzed distillates are presented in Figures S3 and S4. The statistically significant metabolites of these models were determined by assessing the VIP and p(corr) vectors and the two-tailed *t*-test, with thresholds of VIP > 0.5, p(corr) > 0.5, and $p < 0.05$. Tables S3–S8 show the values of these vectors for each metabolite.

The OPLS-DA models between Macedonia–Thessaloniki and Macedonia–Pella samples were not statistically significant ($p > 0.05$), as well as in the case of Crete and Crete-Soultanina samples. This observation could lead to the conclusion that distillates produced from either the same or partially the same grape cultivar, or by a similar production process but in a common region, are difficult to distinguish; however, several metabolites were characteristic for some of these products, as mentioned in Section 3.1.

The responsible biomarkers for the differentiation of Crete and Thessaloniki samples were found to be six ethyl esters and one acetate ester, three alcohols, three flavored volatiles, four terpenes, one acid, and two aldehydes. In the case of Crete and Limnos spirits, four ethyl esters, two flavored compounds, two aldehydes, two acids, five terpenes, and two alcohols were the statistically significant metabolites, while four ethyl esters, four flavored compounds, four terpenes, three aldehydes, two alcohols, and two acids were responsible for the separation between Thessaloniki and Limnos distillates. Moreover, five esters, five flavored compounds, three carbonyl compounds, two alcohols, three terpenoids, and two acids were found to be statistically significant for the model between Macedonia and Limnos distillates, while nine esters, three alcohols, three carbonyl compounds, two acids, three terpenoids, and one flavored compound were responsible for the differentiation between Crete and Thessaloniki samples. The statistically significant metabolites between Crete and Macedonia grape marc spirits were seven esters, three alcohols, three flavored compounds, one acid, four terpenoids, and two carbonyl compounds. Samples of Crete were differentiated from those of Northern Greece, since the climate of Crete is warmer and drier, favoring the cultivation of varieties with less aromatic metabolites; however, they produce spirits with higher alcohol content, explaining the abundance of higher alcohols and carbonyl compounds in these beverages.

Fatty acid ethyl esters, and especially long-chain fatty acid ethyl esters, were found in higher concentrations in Thessaloniki, Macedonia, and Thessaly samples, whilst ethyl isovalerate and ethyl 2-methylbutanoate were more abundant in Crete spirits. An interesting observation is that ethyl lactate was found in higher concentrations in Thessaloniki spirits compared with any other region, and as mentioned in Section 3.1, this metabolite is an indicator of bad fermentation or storage conditions of a spirit or a long-time ensilage. On the other hand, phenethyl acetate, and acetate esters generally, were more abundant in Limnos and Thessaly grape marc spirits, followed by Thessaloniki, while similar behaviors were followed by terpenes and fatty acids. Macedonia is at a higher altitude than the Thessaly plain and this often affects the composition of the grapes and thereby the aromaticity of the produced distillates. On the other hand, Limnos is an island with a Mediterranean climate and volcanic soil, so although all the distillates from these areas were produced from Muscat pomaces, the products showed several differences in their aromatic composition.

Terpenes, as is shown from *alpha*-pinene and 4-carene in Figure 2, were in higher concentrations in Limnos and Macedonia distillates compared with the other regions, while terpenols were more abundant in Macedonia, Thessaloniki, Thessaly, and Limnos spirits. More specifically, hotrienol was found to be characteristic in Limnos samples, while linalool and nerol oxide were more abundant in Limnos and Thessaloniki samples, *alpha*-terpineol and geraniol in Thessaloniki and Thessaly, and terpinen-4-ol in Macedonia and Thessaly samples. Higher alcohols and aldehydes were more abundant in Limnos, Macedonia, and Thessaly samples, produced from Muscat and Muscat of Alexandria grapes, with the exception of 2-methylpropan-1-ol and benzaldehyde, which were detected in higher concentrations in Thessaloniki and Crete spirits (Table S2 and Figure S2).

While numerous factors affect the aromatic profile of grape marc spirits, such as the grape's varietal origin, pomace storage methods, fermentation duration, distillation technology, and characteristics of the barrels used in the aging process [13,14], there is no established relationship between the geographical origin and the volatile composition of tsipouro distillates. Since, in most cases, the distillation process is empirical and based on distinctive local traditions, the influence of the geographic origin can be multifactorial, as in different regions, apart from the unique raw materials, there are different storage, fermentation, and distillation practices that are followed, making it very hard to distinguish the influence of each origin-based variable. In this study, we discussed the observed differentiation among samples from different geographical origins, assessing their aromatic profile, and as in the case of Northern Greece tsipouro samples, whilst they were all produced from Muscat pomaces with a similar production process (addition of anise), they presented great differences in their volatile compositions and especially in their abundances of terpenes and terpenoids, ethyl esters, and fatty acids. The samples of this study were examined as products with a holistic approach, as they were produced from several grape cultivars, with various traditional processes (as many of them are PDO products), and in many different regions. In Greece, it is common for each region to use its own tsipouro production process based on the cultivated variety, the peculiarities of the region, and the market demand. However, this protocol could be suitable for the respective evaluation of the impact of the aforementioned factors.

## 4. Conclusions

The proposed protocol was found to be suitable for the identification of 92 volatile compounds in the analyzed distillates, including alcohols, esters, acids, carbonyl compounds, terpenes, terpenoids, and flavored compounds originating from the addition of anise in spirits. PCA and OPLS-DA models were applied for the classification of the majority of the samples according to their geographical origin, while biomarker assessment revealed the statistically significant metabolites of this discrimination.

An important compound group of the volatilome of Greek grape marc spirits is the group of terpenoids. Linalool, geraniol, and nerol oxide were more abundant in Thessa-

loniki, Limnos, and Thessaly samples produced from Muscat and Muscat of Alexandria pomaces; *alpha*-terpineol and terpinen-4-ol were detected in higher concentrations in Macedonia, Thessaloniki, and Thessaly samples (also produced from Muscat pomaces). All of these samples were aniseed distillates. Tsipouro samples from Macedonia presented high concentrations of monoterpenes, i.e., over 50-fold higher than the other samples in most cases. Limnos samples were found to be richer in hotrienol, anethol, and other flavored compounds. Other compounds, such as fenchone and carvone, were found to be higher in Thessaly and Macedonia samples, while *gamma*-isogeraniol, *trans-p*-mentha-1(7),8-dien-2-ol, and thymyl ethyl ether were very abundant in Thessaly samples.

Regarding the other groups of detected compounds, tsipouro samples from Thessaly presented very high concentrations of fatty acids. Samples from Macedonia were found to be richer in ethyl octanoate, while Thessaloniki, Thessaly, and Pella samples were richer in ethyl decanoate and ethyl dodecanoate; ethyl palmitate was higher in the Limnos and Thessaloniki samples. Most of the higher alcohols were observed in higher concentrations in grape marc spirit from Cyprus, followed by Limnos, Macedonia, and Thessaly samples; 2-methylpropan-1-ol, was more abundant in Crete, Thessaloniki, and Pella distillates.

The results of this study may be a tool for producers in Greece to make high-quality grape marc spirits based on the special aroma characteristics of each production area.

**Supplementary Materials:** The following supporting information can be downloaded at: https://www.mdpi.com/article/10.3390/beverages9030065/s1, Table S1. Sample number, region, grape variety and addition of aromatic material of the analyzed samples. Table S2. Compound name, retention time (RT), chemical formula, molecular weight (MR) and relative concentration (mg L$^{-1}$) according to the production region of the metabolites identified in analyzed Greek distillates; Table S3. VIP, p(corr) and *p*-values of the statistically significant volatile metabolites of the discrimination between Crete-Soultanina and Macedonia Greek grape marc spirits; Table S4. VIP, p(corr) and *p*-values of the statistically significant volatile metabolites of the discrimination between Crete and Limnos Greek grape marc spirits; Table S5. VIP, p(corr) and *p*-values of the statistically significant volatile metabolites of the discrimination between Macedonia and Limnos Greek grape marc spirits; Table S6. VIP, p(corr) and *p*-values of the statistically significant volatile metabolites of the discrimination between Crete and Thessaloniki Greek grape marc spirits; Table S7. VIP, p(corr) and *p*-values of the statistically significant volatile metabolites of the discrimination between Thessaloniki and Limnos Greek grape marc spirits; Table S8. VIP, p(corr) and *p*-values of the statistically significant volatile metabolites of the discrimination between Thessaloniki and Pella Greek grape marc spirits. Figure S1. Boxplots of representative esters among Greek grape marc spirit samples. Black dots represent the samples and yellow dots are the mean values; Figure S2. Boxplots of representative acids, alcohols and carbonyl compounds among Greek grape marc spirit samples. Black dots represent the samples and yellow dots are the mean values; Figure S3. OPLS-DA score plots of the discrimination of (a) Macedonia and Crete-Soultanina, (b) Macedonia and Limnos and (c) Limnos and Crete grape marc spirit samples; Figure S4. OPLS-DA score plots of the discrimination of (a) Thessaloniki and Pella, (b) Thessaloniki and Limnos and (c) Thessaloniki and Crete grape marc spirit samples.

**Author Contributions:** Conceptualization, M.M. and G.T.; data curation, M.M., P.A. and I.S.; formal analysis, M.M. and A.N.; Funding acquisition, G.T.; methodology, M.M., I.S. and G.T.; project administration, G.T.; resources, A.N.A. and G.T.; software, M.M., P.A. and I.S.; supervision, G.T.; writing—original draft, M.M.; writing—review and editing, M.M., P.A. and G.T. All authors have read and agreed to the published version of the manuscript.

**Funding:** This research was supported by the project FoodOmicsGR "National Research Infrastructure for the Comprehensive Characterization of Foods" (MIS 5029057), which is implemented under the action "Reinforcement of the Research and Innovation Infrastructure", funded by the Operational Programme "Competitiveness, Entrepreneurship and Innovation" (NSRF 2014–2020) and co-financed by Greece and the European Union (European Regional Development Fund).

**Data Availability Statement:** All data are included in the main text and the Supplementary Materials.

**Acknowledgments:** We thank local producers for supplying us the grape marc spirit samples.

**Conflicts of Interest:** The authors declare no conflict of interest.

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
