# Peer review of "Analysis of the Volatile Organic Compound Fingerprint of Greek Grape Marc Spirits of Various Origins and Traditional Production Styles"

_beverages, doi:10.3390/beverages9030065_

Round 1
Reviewer 1 Report
I am very grateful you for the invitation to review manuscript beverages-2508087 by Marinaki and coauthors " Analysis of the Volatile Organic Compounds fingerprint of Greek grape marc spirits of various origin and traditional production style”. The aim of this study was to characterize the volatilome of tsipouro products by covering as many geographical origins and production styles as possible. The work is interesting but needs adjustments to increase the quality of the material.
Comments:
- Line 18-20: Specify better the characteristics of the “tsipouro” (alcohol content, etc.).
- Line 34-35: Change the repeated keywords by different words from the title
- Abstract: The “problem” to be solved with the study is not clear. Any specific problem? Lack of characterization of the drink? Please be more specific.
- Line 22: The full denomination must be presented at the first appearance of HS-SPME-GC-MS.
- Abstract: The authors do not present a conclusion about the study.
- Introduction: Highlighting the production and market of “Tsipouro” in Greece and around the world.
- Introduction: A better and more precise definition of the Tsipouro must be presented, including physicochemical and sensorial characteristics.
- Line 96: Change “is to” to “was to”.
- Lines 97-99: This theoretical information should be presented in the introduction, before the objectives to add information about the important properties of beverages.
- Line 114: Change “6of them produced” to “6 of them produced”.
- Lines 121-123: This is referential and should be inserted in the introduction in adequate space.
- Lines 121-130: This information is confusing. Rewrite, showing only the used conditions.
- Line 200: The impact of “geographic origin and distillation techniques” should be better presented, discussing how the process and conditions in the region influence the beverages.
- Results: The discussion regarding the characteristics of the regions and the consequences on the composition must be improved.
- Line 302: Insert the beverage storage time information in the material and methods.
- The discussion about volatile components and their formation is very well presented and detailed. However, authors should include discussions regarding the influence of geographic origins.
- Discussion: Include a brief discussion regarding the importance of characterizing and establishing the indication of origin. In addition, it is important to indicate the ability to detect quality during processing and storage.
- Lines 418-422: Please remove this information. The conclusion should include only the main observations in relation to the objectives of the work.
- Conclusion: Rewrite, succinctly presenting the main observations of the work, related to the objective.
Reviewer 2 Report
A brief summary
The article is interesting with a good potential, and could significantly contribute to the mentioned scientific field as well as applied journal. It is necessary to check the typos and the English language. Everything else is very well explained. However, there are some parts that could be improved before publication.
General comments:
Introduction: This part may be a bit too extensive. The authors describe in detail the methods used by other researchers to determine the volatile composition as well as sample preparation methods, i.e. extraction methods. That part may be unnecessary. Authors should think about whether they want to leave this part or shorten it/delete it.
At the end of the Introduction, there should be only Aim of the study. Remove the part about describing statistics from the Introduction.
Materials and methods:
Chemicals: Are those mentioned standards and chemicals all that were used in this research? If no, please include all standards (mixes or individual) and their purity.
Overall, this Materials and Methods section is written in detail with mostly all the necessary information.
Results and Discussion: Quite well written but contains some information, facts and basics that are not part of the discussion of this study. In this part, it is necessary to stick exclusively to the results of the study and comment them through comparison with other similar studies.
Conclusion: Too long. It needs to be shortened because all the facts mentioned here are not the conclusion of this study.
References: There is a lot of references older than 5 years. Check them and throw them out if they are not necessary, or find a newer one (if possible). Also, since this is not a review paper, perhaps not all of these references are necessary (64!?) Please remove unnecessary ones.
Specific comments:
Line 99-104 This sentence does not belong here. Please remove it to the Methods Section.
Line 107-110 Please rewrite the sentence for better understanding
Line 114 Please correct the typo
Line 150-153 Please write here how much was the total runtime
Line 176-188 This part of the text does not belong to the Results and Discussion Section. Please transfer this text to the Introduction Section or delete this part.
Line 189-195 This text also does not belong in Results and Discussion Section. It is a part of Materials and Methods Section. Please transfer it there.
Line 197-202 Again, this text belongs to Introduction Section because it is not the analysis of the obtained results and their comparison with other authors, which is the point of Results and Discussion Section.
Line 266-267 Correct this sentence - please rewrite it
Line 330-332 Reformulate these two sentences and combine them with the following one. Written like this, it sounds too banal as well as the explanation of basic terms. You could write for example: Acetaldehyde, which is usually formed ....was detected in much lower levels...
Line 354 Include full stop in the end of the sentence
Line 361 Instead starting the sentence with „In Figure 5“ please include something else, like for example: „It is shown...“ or similar. Or simply connect those two sentences.
Line 418-422 This part is not authors conclusion so it should be removed.
The article is well written and understandable but it is suggested to check it with a professional.
Round 2
Reviewer 1 Report
Authors have improved the quality of the work